# Family Business Owner as a Central Figure in Customer Relationship Management

**Siniša Arsić [1],\*****, Koviljka Banjević [2], Aleksandra Nastasić [2], Dragana Rošulj [2] and Miloš Arsić [3]**

1   Department for Management and Specialized Management Disciplines, Faculty of organizational sciences, University of Belgrade, 11000 Belgrade, Serbia

2   Quality Management Department, Belgrade Polytechnic, 11000 Belgrade, Serbia; kbanjevic@politehnika.edu.rs (K.B.); anastasic@politehnika.edu.rs (A.N.); drosulj@politehnika.edu.rs (D.R.)

3   Faculty of Maritime Academic Studies, 11000 Belgrade, Serbia; misaarsa@yahoo.com

*   Correspondence: sinisaars@telekom.rs; Tel.: +381-64-6503-293 or +381-11-211-1453

**Abstract:** This paper presents theoretical and empirical research on the activities and attitudes of a family business owner regarding marketing as a business function. The development of successful business relationships of a family business is tightly connected with the activities of the business owner. The theoretical review examined numerous family and non-family business studies by analyzing the existing paradigms of marketing management as a whole. The empirical research, surveying 420 family businesses in Serbia, defined the overall role of the family business owner in customer relationship management regarding business-to-business (B2B) and business-to-consumer (B2C) relationships. Key findings suggest that the main difference in customer relationship management (CRM) between family and non-family companies is related to B2B relationships, which the family business owner is heavily involved in managing, in terms of invested time and responsibility. Future research should cover aspects of internationalization (to regional markets) because it is essential to cross-examine regional with local contacts of a family business owner, to provide full comprehension of the complexity of market relationships in a family business environment.

**Keywords:** family company; owner; customer relationship management

## 1. Introduction

Marketing is more than just promotion and communication. It is more than a commercial, an advertisement on the Internet or a social network presentation. It encompasses the entire journey of the customer, and it starts from the first touch point (initial purchase) to the moment of contract or service renewal. It represents a wide range of activities from a top-level marketing plan to workflows and customer care. Previous studies on the topic of marketing as a business function in family-owned companies mostly focus on marketing communication [1], branding [2], and product portfolio-related topics [3]. Not many existing studies seem to focus on customer relationship management (CRM) [4–7]. The results of worldwide research on family businesses, conducted by consultant company PwC [8], refer to unpredictable market conditions as one of the major external challenges for the future of family businesses. According to the Annual World Economic Forum report [9], the most important research papers currently trending predominantly concern generation transition, technology and innovation, governance, trust, and the roles of the family business owner, assuming her or him as the main responsible authority for the majority of business decisions.

Studies dealing with the theory and observations of existing and potential trends regarding family businesses have been narrowed, referring mainly to the EU, and then to the Republic of Serbia

specifically. Currently, Serbia is ranked as 48th in Doing Business's world list (2018), having fallen five places in 2018, after moving up 47 places in the previous year. Meanwhile, although PwC [8] reports that 80% of surveyed European family businesses predict steady or rapid growth within the next five years, it remains unknown whether this is the case in Serbia.

In an attempt to examine the current trends in the literature, this paper empirically deals with an important topic in the form of a survey, which was defined with support of expert family business owners (see Appendix A for more detail). The results and discussion sections of this paper attempt to identify similar conclusions, but with clearly analyzed findings which are related to the current state of the Serbian economy [10], based from the family business owners' point of view. The final part of this research paper sums up and announces the future research efforts of the authors.

When examining current knowledge about CRM and family businesses, authors concluded that the majority of the existing research focused on key CRM initiatives in family versus non-family businesses [11], while failing to deeply analyze owner–key partner relationships. Related to previous knowledge about family business management and the role of the owner in managing key customer relationships, the authors needed to address these topics, by defining the following main research question:

**RQ:** *"What is the role of the family business owner, in terms of their level of involvement in key customer relationships in the market?"*

The answer to this question lies within a systematic literature review, and demands conducting empirical research to fully examine dependencies between the family business owner and important stakeholders. It can be claimed that key customer relationships in family businesses have clearly not been analyzed sufficiently in practice, and rarely in theory. Ascertaining the level of involvement of the owner is essential when analyzing key customer relationships, with respect to specific business-to-business and business-to-consumer relationships. In addition, existing highlighted discussions have mostly compared family business relationships in the market to non-family relationships, but there has been little empirical research about the role of the owner involved in these relationships. Furthermore, there has been a lack of quantitative data related to variables of differentiation between family and non-family businesses, and an even greater lack of deep, integrated research about this topic.

Therefore, to compensate for this gap in the literature, the main research objective of this paper is to define and analyze the role of a family business owner in management of customer relationships, by comparing her/his level of involvement in terms of several factors, namely, company size and industry, as well as the type of customer (business or consumer). Firstly, it is needed to theoretically define key differences between family and non-family businesses in terms of key CRM activities, and then measure those differences by analyzing the role of the family business owner. Based on the need to position the owner of a family business in terms of key CRM activities, it is necessary to thoroughly review the existing literature and highlight previous, similar attempts to identify the research gap. The following section presents previous findings on two main topics: key differences in CRM activities of family and non-family businesses and, consequently, the role of a family business owner in managing relationships with business customers and consumers.

## 2. Theoretical Background and Hypothesis Development

By analyzing socio-emotional wealth (SEW) theory, Nunez-Cacho [12] defined family prominence, continuity and enrichment, as the main dimensions which can be derived from the family. On the other hand, Herrero [13] analyzes continuity of family companies within SEW as striving to avoid diversification of ownership, thus centralizing the position of the family business owner. Having that in mind, Tomlinson [14] defines marketing as relationship management between the owner and key customers, building long-term relationships based on trust and quality customer service.

Piercy [15] views CRM as strategic relationships between different business functions, with alignment between supplier and sales management (private consumers and business customers). Existing research [16] examined customer acquisition, loyalty, profitability and retention. The focus of this paper is on the strategic level of CRM, through analyzing the role of the family business owner.

Little difference can be found between strategic planning, budgeting and overall management. Efforts to centralize strategic activities can improve the quality of a strategy by 40% [17]. The main reason behind this measured improvement lies in focusing on customer needs, taking into account the main efforts of the competition. Hardly any profit-oriented business does not express their mission to satisfy their customer base [18].

For the purpose of understanding customer relationship management in a family business context, the authors made an attempt to define the scope of CRM activities, based on existing literature. Okoroafo [19], Bravo [20], and Mandan [21] define customer service and understanding the priorities of the customer as key customer relationship management activities in a family-owned business. To supplement this, it should be said that there is a clear difference between business-to-business and business-to-consumer relations.

### 2.1. Key Differences in CRM Activities of Family and Non-Family Businesses

Court [22] outlines customer-driven marketing (relationship management) as one of four main activities for all companies, and investments of this kind represent a form of organizational attitude [23]. Sometimes the very identification of a business as family-owned can enable greater effects on the market according to measurements made in 26 markets, where the authors found a reported majority of businesses officially referring to themselves as family businesses [24,25]. Hall and Astrachan [26] investigated whether customers prefer to buy from family businesses, and if their branding as a family company helps them attract new customers. Corporate exploitation of the term "family business" lies in boosting trust, contributing to a customer-oriented focus, instead of on the products and services provided to them, which is often forgotten in non-family businesses [27]. Further, good management of customer relationships can even lead to a multi-generational family business spanning centuries, as was analyzed by Astrachan and Astrachan [28]. When it comes to ownership succession (transition), failure to sustain customer relationships may cause the loss of key stakeholders, contributing to a failure of the second generation of business ownership [29]. However, there are a large number of examples when the most painful transition of ownership leads to better management of marketing activities, in the case of a successor who is also a professional marketing manager [30], or in the case of successful integration of family and business management goals [31].

A precondition for forming structured interactions within a CRM system is to step out of individual (one-to-one) relationships initially formed by the family business owner with his most profitable or most loyal customers [32]. The basic assumption is that, as the family business grows in terms of human resources, the owner will tend to delegate the majority of all marketing activities. This should be empirically examined. Following these conclusions, the authors of this paper consider longevity of business relationships [33], transition of ownership [34], and family business branding and trust [35], as key differences in CRM activities between family and non-family businesses. In total, these findings can support the claim that the family business owner is a "game changer" in her/his company, which she/he achieves through centralization of all decisions regarding customer relationships within family businesses.

From the perspective of non-family businesses, when analyzing marketing communication channels, in terms of complexity, a business-to-business (B2B) company often works with vendor partners, resellers or affiliate companies, selling to other businesses through multiple channels [36]. However, a much more important and distinctive feature of B2B relationships is a process that focuses on developing personal contacts within the larger company [37]. On the other hand, private consumers develop emotions for a certain brand, and purchasing decisions can be made based on a personal relationship and sentiment [38]. Many of the best B2B CRM tools have specific functions related to

identifying "gatekeepers" or influential people within a business [39]. By contrast, customer-facing sales is in most cases simple and straightforward—the company is trying to make an approach to persuade him or her to make certain personal purchase options. Hence, all of infrastructure in B2B tools is not necessary for business-to-consumer (B2C) relationships. However, B2C relationships still need their own range of tools aimed at getting the individual customer to make a choice.

To be able to fully explore and elaborate the main research objective, it is necessary to narrow down and analyze the management role of the family business owner. Taking into account measurability and scalability with appropriate quantitative methods and techniques, it was decided that the role of the owner should be analyzed through their level of involvement to test their role in the definition of activities which maximize overall business success. This should justify and validate the main research goal and research question, as well as to ensure a broader perspective in analysis of CRM activities in family companies. Therefore, the main hypothesis can be defined, to provide an answer to the main research question:

**H0.** *"In family companies, key customer relationship management activities with business consumers present the main difference of customer relationship management between family and non-family companies."*

In the remainder of this section, the authors aim to theoretically determine and analyze the role of a family business owner in the definition of successful customer relationship management activities. Also, for ensuring the framework of the empirical part of the research, the authors defined an auxiliary research hypothesis.

### 2.2. The Role of Owner in the Definition of Successful Customer Relationship Management

The success of marketing activities regarding CRM, for enabling the overall success of the company, has been assessed as being very important by the owners of family businesses. Also, most surveyed owners imply that they have sufficient marketing knowledge, compared to a smaller percentage of managers from non-family companies. Additionally, Tokarczyk [40] confirmed through empirical research that the marketing function, as a set of business activities, was exclusively led by the top management of a family company. Mannarino [41] and Tzempelikos [42] found that top management involvement totally mediates the relationship between top management commitment and quality of business relationships with customers. The aforementioned statement positively contributes to financial performance, therefore, flexibility of marketing plans regarding key customers should be further analyzed in terms of adjustability of a family business to eminent (but unexpected) change.

Chen and Popovich [43] define customer relationship management as a "combination of people, processes and technology that seeks to understand the customers of a company. It is an integrated approach to relationship management by focusing on customer retention and relationship development." In practice, there are two approaches to managing customer relationships: business-to-consumer and business-to-business. Business-to-business relationships often consist of supplier– manufacturer relationships or products and services' sales activities. Supplier relationships with family firms were usually more stable, recognizing the importance and significance of the family business owner, as discovered by Urpinen and Safarikova [44]. This fact should be analyzed further. On the other hand, certain studies point out models and links between business performance (in this case, successful customer relationship management) and characteristics such as innovation management [45], as well as internationalization [46]. Second, efforts to internationalize the business are even more cross-related to effective marketing activities, having in mind specific markets and customers present [47]. It must be expressed that internationalization poses a new challenge for many family companies, as it requires a much more specialized approach because the company is stepping out of domestic market boundaries. Moreover, nowadays family businesses must be competitive globally while trying to export their products and services to new markets [48,49].

From all of these conclusions it can be noted that the role of the family business owner is central in defining the right formula for achieving successful relationships with its customers (especially its

key customers), and it should be examined based on a real sample consisting of family businesses. Relationships with business customers (as a supplier/contractor, or as sales to business consumer) tend to be longer than with private consumers; this can correlate significantly with levels of involvement of the owner, so this should be empirically confirmed.

Looking forward to deeper exploitation of the main hypothesis, the authors have defined an auxiliary hypothesis which could help justify the main research question and validate the main hypothesis, H0. The auxiliary hypothesis should provide answers about which marketing activities are directly influenced by the family business owner. After empirical confirmation, the authors will discuss the findings with previous research regarding differences between family and non-family businesses:

**H1.** *"Family business owners are heavily involved in management of key business customer relationships."*

It is essential to express that adequate management of key CRM activities is one of the main preconditions for achieving business goals. Empirical research should confirm if the role of the owner in key CRM activities truly represents the key difference of CRM between family and non-family companies. Analyzing and validating that kind of research can consequently assist in explaining the previously introduced literature gap.

## 3. Methodology

Following the review of existing literature about the topic of marketing management, it can be summarized that the existing literature focuses on subjects such as marketing communication [1], marketing management activities in family businesses, and tools and techniques for realization of business objectives, while a certain number of papers cover industry-specific topics. Meanwhile, previous literature findings observing family businesses focused mainly on succession planning and ensuring continuity (as many as 18%), and topics covering marketing and CRM were researched in only 0.5% of research papers [50].

Having in mind this revealing information, it can be surely legitimate to state that the main subject of this research should be to further exploit topics related to marketing management in a family business environment. More specifically, the authors will first examine the role of the family business owner, to check whether he stands as the "main pillar" of all family business activities. This will be conducted through a short interview with family business owners from various industries. Conclusions formed as a joint product of the literature review and interviews with experts shall be structured within the planned research survey. After the literature review, the authors defined key dimensions which determine the level of involvement (in terms of time and responsibility) of the family business owner in key CRM activities:

- company size, when moving from direct control to integrating specialist officers [51];
- industry sector, differentiating between manufacturing and services and type of product (sole or with renewal and maintenance) [52];
- type of customer—long lasting business or one-time-purchase private consumer [53].

Also, special attention should be focused on previously reviewed key differences between family and non-family businesses, potentially directly influenced by the family business owner. In that manner, key differences can be observed as the following:

- long-lasting relationships between the owner and key customers (business more than consumer) [33];
- transition of ownership does not transition business relationships of the former owner to the latter [34];
- family business branding, relating to products, services, and customer-oriented trust [35].

Having this in mind, the main objective of this study is to define and analyze the role of a family business owner in management of customer relationships, by comparing her/his level of involvement in terms of company size and industry sector, as well as type of customer (business or consumer).

### 3.1. Research Design

As was previously noted, the authors conducted short interviews with experts (family business owners), to fully understand the circumstances in the Serbian business environment, and to better define research survey questions. A total of 10 experts (from various industries, as can be seen in the Appendix A) were asked to cross-analyze their level of involvement, with positioning of key business activities, defined through a resource-based view framework in [54] and [55]. The results of the cross-examination can be seen in Table 1.

**Table 1.** Resource-based view framework with level of involvement.

| Positioning | Involvement of the Owner (Mean Value of 1–5) | St. Deviation |
| --- | --- | --- |
| Price | 4.6 | 0.5 |
| Quality | 4.8 | 0.5 |
| Innovation | 3.6 | 1.5 |
| Service | 4.5 | 1.5 |
| Benefit | 4.1 | 2 |
| Tailored (one-to-one marketing) | 2.55 | 1.2 |

The results of this cross-examination determine that the family business owner is the main pillar of nearly all business and strategic activities (since recorded mean values are high enough, and standard deviation is low enough), and therefore it can be validated that key customer relationship management activities can be analyzed through the role of the family business owner.

### 3.2. Procedure

The empirical part of the research was conducted with a survey (via online questionnaire, available in Appendix B), of a relatively small sample of family businesses in Serbia. Answers to survey questions may help in closing the identified gap in the existing literature. Questions focused on the personal perception of the surveyed family business owner (subjective opinion), along with general trends involving family businesses (objective opinion discussed in a separate section of the paper).

The research questionnaire consisted of 12 questions and three segments. The first part of the questionnaire dealt with general information about the business in order to better classify companies. The second part dealt with business relationships (business-to-business and business-to-consumer), and the third part focused on determining the influence level of the family business owner on key customer relationships. The questionnaire was designed in the closed option format, so that the respondent could choose only one option. Certain questions are designed in the form of a Likert scale, where respondents could assign values from 5 (most important) to 1 (least important).

### 3.3. Sample

The research questionnaire was sent to 7000 addresses, asking family business owners who were doing business in the six most dominant (excluding state-owned companies) industries in Serbia (i.e., the food industry, information technologies (IT), retail sales, transport, machines and parts manufacture, textiles) about key customer relationship management. The response rate was relatively fair, with a total of 420 owners answering all questions. The authors made an attempt to ensure representativeness of the sample in terms of regional (all four regions were represented with similar proportions) and industry aspects (there are examples of family companies across all six of the most dominant industries in Serbia). The overall population of family businesses in Serbia is not defined in the domestic literature, and there is very little formalized data about family companies.

All data generated during the survey was processed by using Stata v.15 (StataCorp LLC, TX, USA). All sources, references and sampled data used in this paper were read and analyzed several times; the data was cross-referenced and tested. The authors based internal validity of the data gathered from the sample on the authenticity of questions. The questionnaire reflects actual real-life processes

in a family business to extract maximum information about the topic from the respondents. Regional representation of surveyed family companies was achieved, through contacting companies across the four main regions of Serbia, accounting for 64%, 20%, 12% and 4% of all companies in Serbia, respectively. Distribution of the sampled companies was as follows:

- 290 companies from the Belgrade region;
- 89 companies from the Vojvodina region;
- 30 companies from the Sumadija and West Serbia region; and
- 11 companies from the South and East Serbia region.

The following section presents results of the empirical research, as well as a proper analysis of research results.

## 4. Results of Empirical Research

### 4.1. Research Findings

Characteristics of the sampled enterprises (regarding general information about surveyed family businesses in Serbia) are presented in Table 2.

**Table 2.** Characteristics of the sampled companies.

| Question | Response | Pct. of Answers (%) |
|---|---|---|
| Number of employees | 1–9 | 40 |
|  | 10–50 | 35 |
|  | 51–249 | 17 |
|  | 250+ | 8 |
| Company type | Services | 39 |
|  | Manufacturing | 34 |
|  | Combined | 27 |
| Presence in different markets? | Only in the domestic market | 42 |
|  | Yes, in Serbian and foreign markets | 33 |
|  | Yes, in foreign markets only | 25 |
| Are the owner and director the same person? | Yes | 53 |
|  | No, the director is a member of the owner's family | 27 |
|  | No, the director is a non-family professional | 20 |

Surveyed businesses operate with other family companies and consumers almost as much as they operate with other businesses (companies and organizations), outlining collaboration with suppliers (contractors) slightly more than through sales of products. There is a significant correlation between family business size and type of business relationship, since it was established that larger family businesses tend to define strong business relationships with just a few business customers (but with more stability across time). This can be a good sign of a family business owner's involvement in management of B2B relationships.

As the previous literature review confirmed, the owner of a family business is a central figure and maintains an essential role in everyday activities of his business. This survey examined levels of involvement of the owner regarding time and responsibility. Definition of product offers and promotion, as well as negotiations and business development, are the most time-consuming activities of the owner. Acquiring new resources is an activity carrying the highest responsibility, according to surveyed owners. Owners are marginally (or not at all) involved in retention of existing customers, and the owners were undivided about this question (S.D. equals 1.25). When focusing on questions about marketing activities definition, the owners were asked if they could rank the priorities between several guidelines. They agreed that in general, the marketing function of the company is coordinating their actions with other departments, which is a sign of autonomy delegated by the owner. Also, the owners noted that deciding on a proper answer to competitor offers in the market presents the main challenge when planning marketing activities.

**H0.** *"In family companies, key customer relationship management activities with business consumers present the main difference of customer relationship management between family and non-family companies."*

As can be concluded from Table 2, the majority of surveyed companies are micro (less than 10 employees) and small (75% with less than 50 employees), operating mostly in the domestic market and providing (managing) services for other people or businesses. Decision making is (equally) divided among family business owners and non-family professionals in cases where non-family professionals are involved in top management.

When asked about the most influential factors in the market, family business owners defined competition and the business environment (which represents the overall aspect including the specific industry, both types of customers, and competition efforts) (Table 3).

**Table 3.** Questions related to marketing activities in a family business.

| Question | Response | Total (%) | Key Customer Relationship Management (CRM) Activities Are: | | |
|---|---|---|---|---|---|
| | | | Under Direct Control of the Owner | Under Control of a Professional Marketing Manager | Outsourced (Marketing Consultant, Agency) |
| Most influential market factors | Competition | 32 | 66% | 11% | 23% |
| | Business environment (industry, customers, competition) | 27 | 44% | 28% | 27% |
| | Customers | 24 | 13% | 40% | 47% |
| | Industry | 17 | 76% | 6% | 18% |
| Company operating mostly with | Business customers (sales of products) | 30 | 55% | 35% | 10% |
| | Business customers, as a supplier (contractor) | 26 | 90% | 10% | 0% |
| | Private customers | 25 | 18% | 15% | 67% |
| | Other family companies | 19 | 89% | 7% | 4% |
| | Total (%) | | 44% | 30% | 26% |

From this table, several conclusions can be made about the majority of relationships:

- B2B relationships, as well as relationships with other family companies, are under direct control of the owner;
- B2C relationships are outsourced to marketing consultants.

These correlations must be analyzed deeper to confirm the statistical significance of these claims. Research variables "company mostly operating with" and "most influential factors of success in the market" were analyzed with a correlation matrix, to determine whether the correlation coefficient between two variables has statistical significance, and also, if B2B and B2C relationships are equally represented in the sample, independent of the variable "most influential factors of success in the market". Analysis of adjusted *R*-squared and correlation coefficient values between research variable data points is displayed in Table 4.

**Table 4.** Correlation matrix within and between variables "most influential factors" and "company mostly operating with".

| | | Private Customers | Business Consumers, as a Supplier (Contractor) | Business Consumers (Sales of Products) | Other Family Companies | Industry | Consumers | Competition | All (ind., cust., comp.) |
|---|---|---|---|---|---|---|---|---|---|
| Private customers | R | 1 | | | | | | | |
| | Sig. | / | | | | | | | |
| Business customers, as a supplier | R | 0.179 | 1 | | | | | | |
| | Sig. | 0.000 | / | | | | | | |
| Business customers (sales of products) | R | 0.166 | 0.116 | 1 | | | | | |
| | Sig. | 0.000 | 0.000 | / | | | | | |
| Other family comp. | R | 0.218 | 0.492 | 0.510 | 1 | | | | |
| | Sig. | 0.000 | 0.000 | 0.000 | / | | | | |
| Industry | R | 0.151 | 0.098 | 0.166 | 0.199 | 1 | | | |
| | Sig. | 0.000 | 0.000 | 0.000 | 0.000 | / | | | |
| Consumers | R | 0.654 | 0.344 | 0.589 | 0.541 | 0.201 | 1 | | |
| | Sig. | 0.000 | 0.000 | 0.000 | 0.000 | 0.000 | / | | |
| Competition | R | 0.582 | 0.155 | 0.549 | 0.113 | 0.293 | 0.325 | 1 | |
| | Sig. | 0.000 | 0.000 | 0.000 | 0.000 | 0.000 | 0.000 | / | |
| All (ind., cust., comp.) | R | 0.158 | 0.479 | 0.452 | 0.345 | 0.203 | 0.244 | 0.511 | 1 |
| | Sig. | 0.000 | 0.000 | 0.000 | 0.000 | 0.000 | 0.000 | 0.000 | / |

R—Residual; Sig.—significance; ind.—Industry; cust.—Consumers; comp.—Competition.

It can easily be seen that the highest correlations are between the following variable combinations:

- companies dealing mostly with private consumers, considering them the most important factor on the market (correlation coefficient R equals 0.654);
- companies dealing mostly with business customers (sales) and private consumers, regard their competitors as the most influential factor on the market (correlation coefficients are 0.582 and 0.549, respectively);
- companies dealing mostly with other family companies, hold that customers are the most important market factor (R equals 0.541), while individual competitors are not relevant (R equals 0.113).

It can also be noted from Table 4 that the variable "industry" is statistically insignificant in terms of correlation when considering dependence between variables, since there is obviously a lot of variability in terms of the industry in which surveyed family businesses are located. Hence, from the fact that the owners rated this variable with a very poor score in the correlation matrix it can be concluded that it does not contribute to testing the main hypothesis, but this revelation helps in reaching the main objective of this research.

The results of correlation tests between variables "company mostly operating with" and "most influential market factors" are displayed in Table 5, and it can be confirmed that there is little variability between the derived correlation matrix (determined correlation coefficient) and the adjusted *R*-squared test. Overall variability in all research variables is between 0.78 and 0.72, which is a good confirmation of sampled data. Validation tests will follow at the end of this section.

**Table 5.** Correlation tests and adjusted *R*-squared.

| Variable Test | Company Mostly Operating With | Most Influential Market Factors |
|---|---|---|
| Cor (x,y) | 0.73 | 0.72 |
| Adjusted $R^2$ | 0.78 | 0.74 |

Conducted tests (linked with Tables 3–5) are sufficient to confirm the main hypothesis, but for the overall process of defining and analyzing the role of a family business owner in management of customer relationships, further testing and comparations are required, through testing of the auxiliary hypothesis:

**H1.** *"Family business owners are heavily involved in management of key business customer relationships."*

Results of cross-referencing questions about who controls marketing activities with the level of involvement (in terms of time and responsibility), clearly support proper testing of the auxiliary research hypothesis, and are presented in Table 6. CRM activities are in most surveyed cases (44% of companies) under direct control of the owner, which is a sign that the owner is essential for activities related to the execution of the marketing strategy. This claim should be tested statistically for validation.

**Table 6.** Cross-referenced data on level of involvement over key CRM activities.

| Key CRM Activities | Level of Involvement of the Owner | Key CRM Activities Managed By | | | Mean |
|---|---|---|---|---|---|
| | | Separated as a Function | Owner Controlled | Outsourced | |
| Offering and promotion definition | time wise | 3.54 | 3.17 | 2.13 | 3.26 |
| | responsibility wise | 3.04 | 2.86 | 2.76 | 3.09 |
| Negotiations and business development | timewise | 3.08 | 3.06 | 3.03 | 3.05 |
| | responsibility wise | 3.13 | 3.15 | 3.26 | 2.51 |
| New market exploration | timewise | 2.93 | 3.31 | 3.07 | 3.09 |
| | responsibility wise | 2.89 | 3.23 | 3.04 | 2.39 |
| Retention of existing customers | timewise | 2.25 | 3.00 | 2.46 | 2.51 |
| | responsibility wise | 2.17 | 2.82 | 2.78 | 2.87 |
| Acquiring new resources | timewise | 2.43 | 1.80 | 2.64 | 2.39 |
| | responsibility wise | 2.58 | 3.00 | 2.35 | 3.19 |
| Timewise total | | 2.85 | 2.87 | 2.86 | |
| Responsibility wise total | | 2.76 | 3.01 | 2.84 | |

From Table 6, several conclusions can be made about the opinion of the surveyed owners. The owner directly manages key CRM activities. Time-wise and responsibility-wise, his/her involvement level falls to acquiring and retaining existing customers, as well as to exploring new markets. Other than that, the owners who decided to delegate "offerings and promotions definition" (time-wise) to marketing consultants, were still very involved in "negotiations and business development" in terms of responsibility. To be able to confirm the auxiliary hypothesis H1, the authors had to test the following variables:

- Involvement level of the family business owner,
- Control level of marketing activities.

Firstly, the authors conducted a *t*-test to examine the difference between mean variables. Since there was initial significance (based on *p*-value < 0.05), it was decided to conduct further tests. An established regression between these two variables would support the claims about the correlation between rising level of involvement and management of key CRM activities. Accuracy of the auxiliary hypothesis will be confirmed by performing a multiple linear regression with 95% confidence interval, by obtaining values of the *F* statistic, $R^2$ and residual standard error (RSE).

Equation (1) shows the derivation of the *F* statistic, based on the observed total sum of squares (TSS), the residual sum of squares (RSS), the number of variables (*p*), and the number of observations (*n*). High values of the *F* statistic (significantly greater than 1) shall confirm the relationship presented in the first auxiliary hypothesis, bearing in mind that the value of *n* (i.e., the sample size) is relatively small. Additionally, the authors used the adjusted *R*2 statistic, which increases only if a new variable improves the regression more than is expected by chance.

$$F = \frac{(\text{TSS} - \text{RSS})/\text{p}}{\text{RSS}/(\text{n} - \text{p} - 1)} \tag{1}$$

Values of the adjusted $R^2$ statistic near 1 signify a strong correlation between the variables "key CRM activities are managed by" and "involvement level of the owner"; in other words, how much variability in data around the variable "involvement level of the owner" is directly explained by (correlated with) the variable "key CRM activities are managed by". The results of the conducted tests are displayed in Table 7.

**Table 7.** Regression test results (confirmation of auxiliary hypothesis).

| Test Variable | Adj $R^2$ | F | Residual Standard Error (RSE) | *p*-Value |
|---|---|---|---|---|
| involvement level of the owner | 0.75 | 440 | 1.65 | 0.002 |
| key CRM activities are managed by | 0.88 | 355 | 1.48 | 0.003 |

Based on analysis of the results of the four conducted tests there is a significant relationship between the variables (adjusted $R^2$ is very close to 1, $F$ statistic values are much larger than 1, RSE is slightly above the standard deviation level, and the *p*-value is smaller than 0.05) and it can be concluded that first auxiliary hypothesis is confirmed, or, family business owners are heavily involved in management of key B2B activities in terms of time and responsibility.

*4.2. Validation of Research Results*

Displayed research results were validated with separate tests, on both main and auxiliary hypotheses. Based on sample size and complexity of the sample (several dimensions), it was most appropriate to use a "leave one out cross validation" test (LOOCV), by analyzing the variability ($MSE_i$) in estimating $F$ statistics and adjusted $R^2$. The main intention was to avoid bias in estimation of the mean squared error, or in other words, it was necessary to achieve a satisfactory level of variability across the sample (this test aimed to measure variability between owners' opinion). Equation (2) shows the calculation for the cross validation test, and Figures 1 and 2 provide a graphic display of LOOCV tests on a sample of 420 family businesses. The graphic display of the LOOCV sample results ($n = 420$, one dot represents the five nearest results), in terms of $F$ statistic and adjusted $R^2$ determinants, are displayed in Figures 1 and 2 respectively.

$$CV_{(n)} = \frac{1}{n} \sum_{i=1}^{n} MSE_i \tag{2}$$

It can be concluded that there is enough variability across the sample, measured through the $F$ statistic and adjusted $R^2$. Variability determined with LOOCV ensures that the $F$ statistic and adjusted $R^2$ are statistically significant, and it is safe to claim that there is no bias under which the owners formed their responses, which would result in false conclusions and empirically unsupported claims.

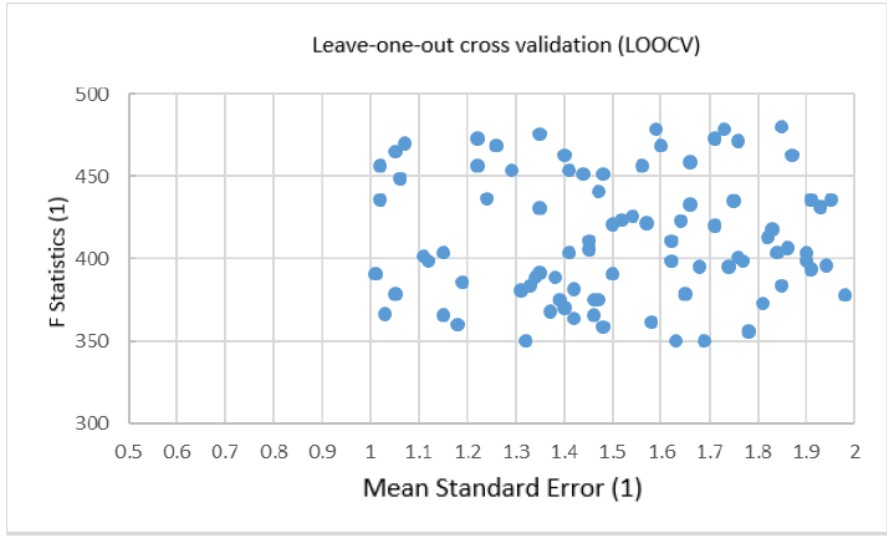

**Figure 1.** Leave one out cross validation test referring $F$ statistic to Mean standard error (MSE).

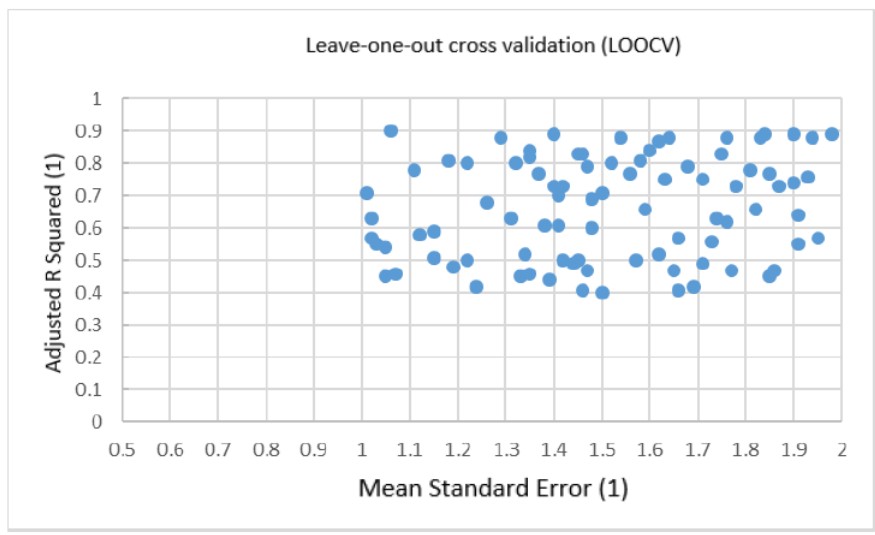

**Figure 2.** Leave one out cross validation test referring adjusted $R^2$ to MSE.

## 5. Discussion

Chrisman [56] discovered that too much involvement of the family members throughout the business can endanger its performance, but not clearly outlining the role of the owner. Maury [57] introduces family control as a balance mechanism between owners and managers (if they are not the same person). A study by financial consultants KMPG [58] supports these claims by stressing the importance of family-business balance. This paper identified the role of the owner as the overall manager of key customer relationships, but with clear differentiation based on customer type. Research presented here covered these aspects with more detail, by observing relationships in B2B and B2C segments, covering the gap discovered by Hall and Astrachan [59], who did not differentiate CRM based on the type of customer. Also, Jacobson [60] focused his research on identifying differences between B2B and B2C regarding customer retention, while this research contributed with new correlations between the most influential factors on the market and the type of customer (business relationships).

Stable and sustainable growth of a family business usually involves organized marketing efforts, but under direct guidance of the owner. This fact was confirmed in the research, and it can be concluded that it presents one of the preconditions for successful implementation of the marketing strategy in a family business [61–63].

This paper did not consider trends concerning external factors, such as integration of social responsibility with marketing activities of organizations [64], family members employed in the company and motivation factors, both influencing the owner, in efforts to build business results [65]. In the case of an emerging market like Serbia, circumstances where corporate identity and government regulations do not guarantee enforcement of deals and agreements, personal commitment from the owner of a family business can be as powerful as a signed contract, according to a research from McKinsey professionals [66]. This paper confirms that claim, discovering that the owner is directly in charge of negotiations and business development. Memili [67] suggests that family owners and managers may have more or less power than their peers in different countries, so the level of centralization in CRM of family companies must be examined further with international cross-cultural understanding.

All issues empirically covered by this research should be analyzed by involving a longer period for observation, to ensure that rapid changes of market structure and circumstances in evolving markets (such as in the case of Serbia) do not create too much "noise" in generating quality conclusions. Until the expected expansion of this research, conclusions made here cannot stand as proof that family business owners are constantly behaving in the presented manner. Nevertheless, this research

represents a good prelude for further analysis of the identified literature gap and it has the potential to determine the course of future studies.

## 6. Conclusions

The theoretical review concerning marketing CRM and family business studies integrally examined the role of owners in the creation and execution of successful customer relationship management activities, taking into consideration specific industries and the parent (regional markets) where a family company operates. The key contribution of this paper is the recognition of actual trends in the implementation of CRM activities, defining the role and level of involvement of the owner. The second most important result of this paper is empirical research of family businesses in Serbia, examining the realization of customer relationship management activities in family businesses overall. The owner (as the research sample suggests) plays a significant part in providing support for the abovementioned processes, and he directly influences key business customer relationships, unlike in non-family businesses.

When considering restrictions for proper research on this topic, there is a clear fact that the business environment in terms of family-owned companies hasn't been explored fully in Serbia or overall. Still, the main limitation of this research is a failure to explore surrounding markets (Central European Free Trade Association, or CEFTA region), in terms of key CRM activities of family businesses. The owners largely agree that actual trends evidence change of market share and internationalization (import/export activities) of their business. Further research on regional business relationships should be done by following guidelines defined by Kontinen [68], as well as by Sist [69].

There is a clear gap left for further research since there is a preference of the EU for further expansion and integration, at least in terms of business relationships with West Balkan countries [58]. This paper represents the first step in the process of analyzing the role of the family business owner in the management of key CRM activities. Further research should further emphasize the importance of a family business owner's (and his/her family's) involvement in activities which maximize business success. Another potentially important contribution of future research is to provide and present more knowledge about family businesses in Serbia, their management structure and marketing proceedings, as well as to describe best business practice examples from Serbia as a developing country with rapid, but sustainable growth.

*Practical Implications*

From a practical point of view, results of the empirical research conducted within this manuscript present a good example of how much of the influence on the business outcomes of the family business is directly related to efforts of the family business owner. This study contributes to existing literature (with clearly displayed limitations and restrictions), by investigating the role of the family business owner, in terms of level of involvement in key business customer relationships. The main presumption (investigated through main and auxiliary hypotheses) was that CRM of family differs from non-family businesses, mostly as the result of specific relations with B2B customers, with heavy involvement invested by the family business owner. The main objective of this study was to define the mentioned business customer relationships of the owner, by comparing her/his level of involvement in terms of company size and industry sector, as well as type of customer (business or consumer). Analysis of empirical research results reveals that industry sector and company size are not relevant (statistically significant) when analyzing the level of involvement of the owner, which is similar to conclusions made by Cooper regarding these dimensions [11].

While surveyed owners revealed that marketing activities in their company are delegated to a specific marketing business function or marketing consultants, they also strongly underlined their level of involvement in market relationships with business customers and other family companies, in terms of invested time and responsibility. The owners largely agree that offering and promotion definition for key customers is the most time-consuming activity, and also that acquiring new resources

requires the highest level of their involvement in terms of responsibility. Yet it remains unclear whether the owners would even allow (and under what circumstances) another outcome to take place, in terms of level of involvement.

These relationships are depicted as maintaining longer lasting relationships with other family companies, or with business customers, but this research also revealed that the owners are highly involved in new business customer acquisitions. This can be a good indicator of how the owner balances stability of existing revenues and business contracts, while at the same time is highly responsible for establishing new sources of business opportunities in the market. After analyzing the owners and the influences of current overall business conditions in Serbia as a country in development, there are two main descriptions derived from this research: it is a highly competitive market with frequent changes, and it has an emerging growth trend in internationalization of products and services. This correlates again with the role of the family business owner, whose strategic actions for doing business are led by investing in customer acquisition (retention), and by defining proper responses to competition offers. All of these mentioned factors can be understood as part of an ongoing effort of the owner to diversify risks of losing key business customers, and as a way of behavior within a highly competitive market ecosystem, influenced both from the family and non-family companies' side.

**Author Contributions:** Each author has participated and contributed sufficiently to take public responsibility for appropriate portions of the content. Conceptualization, S.A. and M.A.; Investigation, K.B. and A.N.; Formal analysis, Methodology and Validation, S.A. and D.R.; Project administration and Supervision, A.N.

**Funding:** This research received no external funding.

**Conflicts of Interest:** The authors declare no conflict of interest.

## Appendix A

**Table A1.** Statistics about interviewed experts.

| Industry | No of Experts | Family Business Size (empl. number) | Years of Doing Business | Region of Serbia |
|---|---|---|---|---|
| food industry | 2 | 100+ | 50+ | Vojvodina |
| IT | 2 | 80+ | 10+ | Belgrade, Sumadija and Western Serbia |
| retail sales | 1 | 500 | 25 | Eastern Serbia |
| transport | 1 | 3000 | 30 | Belgrade |
| machines and parts manufacturers | 2 | 20+ | 40+ | South Serbia and Vojvodina |
| textile industry | 2 | 100+ | 50+ | South and East Serbia |

## Appendix B

**Table A2.** Basic information about sampled companies.

| Question | Variable Description | Response | 1–9 | 10–50 | 51–249 | 250+ |
|---|---|---|---|---|---|---|
| Number of employees | Cathegorical variable relating to company size | Percentage of answers (%) | 40 | 35 | 17 | 8 |
| Firm type | Nominal variable relating to sector of business | Response | services | manufacturing | | combined |
| | | Percentage of answers (%) | 39 | 34 | | 27 |
| Are the owner and director the same person? | Nominal variable, only one option can be selected | Response | yes | No, the director is a non-family professional | | No, the director is a member of the owner's family |
| | | Percentage of answers (%) | 33 | 40 | | 27 |

**Table A3.** Regional representativeness of sampled companies.

| Region | Overall Number of Companies | Overall Number of Family Companies | Sample Used in this Research |
|---|---|---|---|
| Belgrade region | 210,000 | 30,000 | 290 |
| Vojvodina region | 75,000 | 7,000 | 89 |
| Sumadija and Western Serbia region | 40,000 | 5,000 | 30 |
| South and Eastern Serbia region | 15,000 | 2,000 | 11 |
| Republic of Serbia | 340,000 * | 44,000 * | 420 |

* estimated.

**Table A4.** Information about marketing relationships and activities.

| Marketing Activities are: | Response | Under Control of Marketing Manager | Under Direct Control of the Owner | Outsourced (Marketing Consultant, Agency) | |
|---|---|---|---|---|---|
| | % of total | 30 | 44 | 26 | |
| Most influential factors on the market: | Response | industry | consumers | competition | Business environment (industry, customers, competition) |
| | % of total | 17 | 24 | 32 | 27 |

**Table A5.** Information about longevity of major relationships of the owner.

| | | Business Consumers | | | |
|---|---|---|---|---|---|
| Company Operating Mostly With | | Other Family Firms | As A Supplier (Contractor) | Sales Of Products/Services | Private Consumers |
| Longevity of major relationships (%) | Less than a year | 20 | 24 | 35 | 68 |
| | 2–5 years | 36 | 29 | 32 | 27 |
| | Longer than 5 years | 44 | 47 | 33 | 5 |
| | % of total | 30 | 26 | 25 | 19 |

**Table A6.** Information about the positioning of the family business owner in CRM.

| Question | Variable Description | Answer | Standard Deviation | Average Value (Mean) |
|---|---|---|---|---|
| Level of influence of factors on the current business condition in the company | ordinal values 1—least important; 5—most important) | Market share has changed | 1.48 | 3.19 |
| | | Products and services have been internationalized | 1.49 | 3.14 |
| | | Merger/acquisition occured | 1.38 | 3.06 |
| | | Upsell of products and services sourced from innovation | 1.43 | 2.79 |
| Definition of involvement level (timewise) in everyday functioning of the company | ordinal values (1—least important; 5—most important) | Offering and promotion definition | 1.62 | 3.26 |
| | | Negotiations and business development | 1.50 | 3.05 |
| | | New market exploration | 1.32 | 3.09 |
| | | Retention of existing customers | 1.25 | 2.51 |
| | | Acquiring new resources/sources of capital | 1.44 | 2.39 |
| Definition of involvement level (responsibility-wise) in everyday functioning of the company | ordinal values (1—least important; 5—most important) | Offering and promotion definition | 1.32 | 3.09 |
| | | Negotiations and business development | 1.25 | 2.51 |
| | | New market exploration | 1.44 | 2.39 |
| | | Retention of existing customers | 1.44 | 2.87 |
| | | Acquiring new resources/sources of capital | 1.64 | 3.19 |
| How do You position the main strategic framework of Your company? | ordinal values (1—least important; 5-most important) | By investing in customer acquisiton and retention | 0.44 | 3.93 |
| | | By defining proper response to competition offers | 0.75 | 4.20 |
| | | By consulting internally and gathering new insights | 1.47 | 2.45 |
| | | By innovating to disrupt present market conditions | 1.37 | 2.97 |

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
