# Peer review of "Family Business Owner as a Central Figure in Customer Relationship Management"

_sustainability, doi:10.3390/su11010077_

Round 1

Reviewer 1 Report

The paper entitled “Family business owner as a central figure in customer relationship management” presents a paper with an interesting and relevant research topic. Still some aspects can be improved:

1.      In the introduction, part authors should state more clearly the research gap regarding the subject of their research, not regrading current state in Serbia. They should emphasize more clearly why it is necessary to analyze the role of business owner in CRM – state more clearly what we know, what we don’t know and what is our contribution based on all this.  The introduction itself should already indicate research gape

2.      It is not usual to say Main objective: and then state – everything should be written as sentence

3.      In literature review, part 2., authors explain scope of CRM; and difference between B2B and B2C relations; then again in 2.1. they define and explain difference among these two types. It is not clear why they repeat this. Especially as 2.1 is about “The role of owner in the definition of successful customer relationship management” so this is the subject that should be explained here, not something else. This is a part where a more critical overview of theoretical concepts and previous empirical research on the role of owner in successful customer relationship management should be analyzed and presented.

4.      In general is it necessary to emphasize and explain in details B2B and B2C relations – especially as this is not the main focus of the paper?

5.      I believe that in the text it would be more appropriate if first CRM of family and non-family business is explained, and then to present the role of family owner. To me this would make more logical sequence of thoughts

6.      In part 2.2 – are there any existing studies on this subject – on the difference of CRM of  family and non-family businesses? In my opinion authors need to additionally provide some specifics of CRM of non family business – currently it is all about one side – the side of family business- Incorporating both perspectives would provide a much clearer picture

7.      In part 3. authors state „review of existing literature about the topic of marketing management, it can be summarized that existing literature focuses on subjects such as marketing communication, marketing management activities in family businesses….“ - Please provide reference for each type of subjects you mention

8.      Authors state: “After the literature review, the authors defined key dimensions which determine the level of involvement of the family business owner in key activities“ – in which literature review? If you use this than this should also be presented in your paper – provide this literature review

9.      As related to hypothesis - I personally believe it would be much better if authors had a part entitled Theoretical background and hypothesis development and then provide theoretical background focused on each hypothesis and then present their hypothesis. In that way they provide theoretical background and reasoning for each hypothesis, reflecting previous studies and gaps in the current literature related to their hypothesis

10.  Authors state - research shall be conducted – it is not usual to write in future tens

11.  Please reconsider is it really necessary to have Table 1?

12.  It is not usuall to present research limitations in the results – limitations should be at the end of the paper

13.  It is really not clear how the conducted tests are sufficient to confirm the main hypothesis? This is very questionable as your test and items used could not reveal that CRM activities in family companies are designed to achieve success on the market, equally in relations with B2C as well as with B2B customers

14.  I do not see any contribution of you correlation matrix – Table 4

15.  In general I do not see any contribution of your H0 – as the main focus of your paper is the one encompassed by H1 and this is something you need to focus on. In my opinion leave out H0 from your paper

16.  In the Discussion part the first paragraph is not necessary – there is no need to explain what is discussion

17.  Authors state: „The paper clearly highlights the shortcomings of the existing literature“ – I did not get this impression after reading your paper

18.  Limitations od the study and future studies is something that should be in the conclusion

19.  Authors need to provide additional theoretical and practical implications

Author Response

Please find our detailed reply in the attached word document.

Best regards,

team of authors

Reviewer 2 Report

The Article Should be considered for publication in another Journal (on family business, marketing or similar). The article does not responds to the areas of interest of the Sustainability journal. 

Author Response

Please find our reply to Your review in the included attachment (word doc).

Best regards,

team of authors

Reviewer 3 Report

REVIEW OF:

Family business owner as a central figure in customer 2 relationship management

INTRODUCTION

The section is well developed. It could be useful to remark what we know and what we don’t know about CRM and FBs. Certainly there is a clear gap that the author want to fill, so they need to stablish this clearly, including references about the topic.

It could be useful to add paper about FBs of Sustainability such as F.J. “Family Businesses Transitioning to a Circular Economy Model: The Case of “Mercadona”. Sustainability 201810, 538”and paper about CRM such as “Customer relationship management: A comparative analysis of family and nonfamily business practices. MJ Cooper, N Upton, S Seaman - Journal of Small Business …, 2005”

LITERATURE REVIEW

To my view, it could be better to include a theoretical framework that explains why the owner influences on the CRM, perhaps Stewardship theory, familiness or SEW.

I think that the hypotheses are better placed in this section, not in method, which Is more numerical section, not theoretical. The hypotheses are derived from theoretical arguments. So, I would placed Ho after 2.2 section, and H1 after of 2.1 section.

So,

First, literature and theoretical framework. From here 2.1 and H1, and after that 2.2 and H0, then methods section.

The conclusion of the Literature review (page 4, line 175 and ss) may be placed on the second section, and here just the explanation of he methods. It could be useful utilize subsections such as: Sample, Research design, and Procedure.

On page 5 authors state:

….to cross-analyze their level of involvement, with positioning of key 226 business activities, as it was defined before, in the resource based view framework [34]. The results 227 of the cross-examination can be seen in Table 1. below….”

Where is this definition and RBV framework? Why RBV appear here?

RESULT

In this section appears the information of sample. This could be moved to methods better, and after the results

The discussion and conclusion is well developed, perhaps it could be interesting explain how the result give the answer to the research question

Author Response

Please find our detailed reply to the comments, in the attached word document.

best regards

team of authors.

Round 2

Reviewer 1 Report

Authors have accepted majority of the comments, altghough some could be additionally done -  for instance it was stated in the first report that  " In part 3. authors state „review of existing literature about the topic of marketing management, it can be summarized that existing literature focuses on subjects such as marketing communication, marketing management activities in family businesses….“ - Please provide reference for each type of subjects you mention" -  Authors have provided for the first, but not for other subjects they mention.

In general,if other reviewers find it suitable for publication, it si also fine by me.

Reviewer 3 Report

Good work, congrats